# AgriSecure: A Fog Computing-Based Security Framework for Agriculture 4.0 via Blockchain

**Sasmita Padhy [1], Majed Alowaidi [2,*], Sachikanta Dash [3], Mohamed Alshehri [2], Prince Priya Malla [4], Sidheswar Routray [5,*] and Hesham Alhumyani [6]**

1   School of Computing Science and Engineering, VIT Bhopal University, Bhopal-Indore Highway Kothrikalan, Sehore 466114, Madhya Pradesh, India
2   Department of Information Technology, College of Computer and Information Sciences, Majmaah University, Majmaah 11952, Saudi Arabia
3   Department of Computer Science and Engineering, GIET University, Gunupur 765022, Odisha, India
4   School of Electronics Engineering, Kalinga Institute of Industrial Technology, Bhubaneswar 751024, Odisha, India
5   Department of Computer Science and Engineering, School of Engineering, Indrashil University, Rajpur, Mehsana 382740, Gujarat, India
6   Department of Computer Engineering, College of Computers and Information Technology, Taif University, Taif 21944, Saudi Arabia
*   Correspondence: m.alowaidi@mu.edu.sa (M.A.); sidheswar69@gmail.com (S.R.)

**Abstract:** Every aspect of the 21st century has undergone a revolution because of the Internet of Things (IoT) and smart computing technologies. These technologies are applied in many different ways, from monitoring the state of crops and the moisture level of the soil in real-time to using drones to help with chores such as spraying pesticides. The extensive integration of both recent IT and conventional agriculture has brought in the phase of agriculture 4.0, often known as smart agriculture. Agriculture intelligence and automation are addressed by smart agriculture. However, with the advancement of agriculture brought about by recent digital technology, information security challenges cannot be overlooked. The article begins by providing an overview of the development of agriculture 4.0 with pros and cons. This study focused on layered architectural design, identified security issues, and presented security demands and upcoming prospects. In addition to that, we propose a security architectural framework for agriculture 4.0 that combines blockchain technology, fog computing, and software-defined networking. The suggested framework combines Ethereum blockchain and software-defined networking technologies on an open-source IoT platform. It is then tested with three different cases under a DDoS attack. The results of the performance analysis show that overall, the proposed security framework has performed well.

**Keywords:** IoT; blockchain; SDN; precision farming; cyber security; fog computing

## 1. Introduction

Farming is the main source of food and contributes significantly to the economy. According to the FAO (Food and Agriculture Organization of the United Nations), by 2050, the demand for producing food must be increased by 70% to satisfy the world's demand. In recent studies, 550 million people of the world's population are in nutrition deficiencies; every day, around 820 million people are still not obtaining food. The study says that the world's population will increase to more than two billion, and most of the population will live in cities. It has also been observed that the population in India and Nigeria will increase to around 482 million between 2019 and 2050 [1].

The biggest challenging situation arises with this population growth to achieve the Sustainable Development Goals (SDGs) of zero hunger [2]. These projections for the near future have an impact on overall food demand. Because of the scarcity of water today, it is

difficult to meet only 40% of the water supply by 2030, and the unavailability of farming land will lead to a reduction in the food supply. As a result, the agriculture sector needs sufficient resources than are presently available, as well as more self-sustaining systems to boost cultivation percentages while reducing environmental resource use [3]. Though it might be appropriate to reach rising demand, it is unclear how to do so in an equitable and resilient manner. Again, there is a vital need to accelerate and scale up the agricultural production transformation [4]. Agriculture 4.0 can provide a way to increase agricultural efficiency with the available crop area. It also optimizes irrigation to use limited water and energy and allocate resources to protect crops effectively. Those would be possible through the integration of environmental monitoring, forecasting, and smart devices [5]. Smart farming's new technologies can enhance agricultural mechanisms, allowing cultivation to increase while using natural resources efficiently. Agricultural sector innovations are referred to as the "digital agricultural revolution", which turns most factors of agriculture, ensuing in more effective, efficient, self-sustaining, integrated, truthful, and self-reliant agriculture. Cellular devices, sensors, data analysis, fog computing, information security, and intelligent systems influence how technologies are integrated into the agricultural sector [6].

Nowadays, agriculture is changing quickly as it enters a new age known as Agriculture 4.0. Agriculture 4.0 seeks to employ new technology and approaches to address the issues facing modern agriculture (such as climatic alteration, illnesses, extreme application of resources, etc.) and to lower risks while enhancing production efficiency and safety [7]. To do this, it makes extensive use of cutting-edge ICTs.

In addition to this evolution, there is a rising trend in food needs. Given the continual evolution and the rising need for food, it is anticipated that the market for agriculture 4.0 will expand greatly in the years to come.

Wireless Sensor Networks (WSNs) and Internet of Things (IoT) technologies are widely utilized in Agriculture 4.0 and provide farmers with a number of advantages, including observing numerous environmental factors connected to crops, spotting crop illnesses, predicting yield, and lowering labor costs [8]. However, the Sensor and network connectivity and communications in farming might act as a base factor for many types of attacks since these devices frequently have obsolete or unpatched firmware or software [9].

Security is a top priority in agriculture, and any interruption or distortion may present difficult problems and have disastrous outcomes. Monitoring and categorization of network traffic, which have attracted a lot of attention since the earliest beginnings of the Internet, can be crucial to preventing network assaults [10]. Scientific research on network traffic categorization for IoT system security has been extensive. It is an essential part of intrusion detection systems (IDS) and aids in the identification and detection of harmful network activity.

This revolution in agricultural technology is built on having access to all the data produced by every sensor used on a farm, centralized via the internet, and analyzed for the best possible decision-making [11]. Any sensor employed generates a significant amount of data that can be extensively examined and provides useful information to enhance crop quality or lessen environmental impact. Cloud computing, physical hardware, and the IoT are the three key components that enable the automation and data sharing needed by this new smart agriculture [12]. With the help of technical improvements (drones, sensors, software, etc.), we can complete agricultural jobs faster and more effectively than humans.

Agriculture 4.0 must make sure that an appropriate security mechanism is implemented to avert an attack to build scalable and secure systems. The integrity of data is important for the effective functioning of information technological advances such as data analysis and intelligent devices [13]. Because agriculture 4.0 incorporates elements from conventional Internet, cellular, and wireless technologies, which may include all of the security risks that advancements present situation. It also addresses current security risks such as data and device integrity, consistency, and accessibility. Devices in agriculture 4.0 are implemented in outdoor spaces where they are subjected to external factors such as ani-

mals, human beings, or farming equipment. Unknowingly, these factors remove or damage sensors. Since the 6th century B.C., there has been a threat known as Agro terrorism [5]. This type of terrorism could have a variety of goals, including causing financial harm, fear, and social unrest [6,7]. Terrorists could incite community disturbance and lack of credibility in administration by exploiting agricultural and food corporate crises, which can lead to undivided attention in the world. Cyber-agro-terrorism in agriculture 4.0 is an electronic system in agricultural settings to harm crops and animals while causing economic loss. Cyber-agro-terrorists can operate attacks on fields and through the internet, using cyber resources [8].

## 2. Overview of Agriculture 4.0

Agriculture has gone through many revolutionary movements, all of which have increased the agriculture industry's performance and effectiveness.

Plant cultivation "in 10,000 BC" resulted in the birth of the world's first societies and civilizations. Machines were used in the farming sector to execute work, which increased production from 1900 to 1930. Farmers were able to use newly discovered crops and agrochemicals during the Green Revolution (from around the 1960s). Biotechnology facilitated the production of plants with previously selected attributes, with increased yield and pest resistance, desertification, and herbicide, from 1990 to 2000. Now, in the 21st century, digitalization has the potential to help humanity survive and flourish for a long time [4].

The world's largest industry, agriculture, contributes significantly to both social stability and economic growth [13]. A growing number of studies on smart agriculture are driven by the task of resolving the conflict between the population increase and the constrained food output. The advancement of science and technology fuels the revolution in agriculture, which is based on both the rise in output and the limitations of the time [14]. Figure 1 is used to explain the features of agriculture development to readers (from Agriculture 1.0 to Agriculture 4.0).

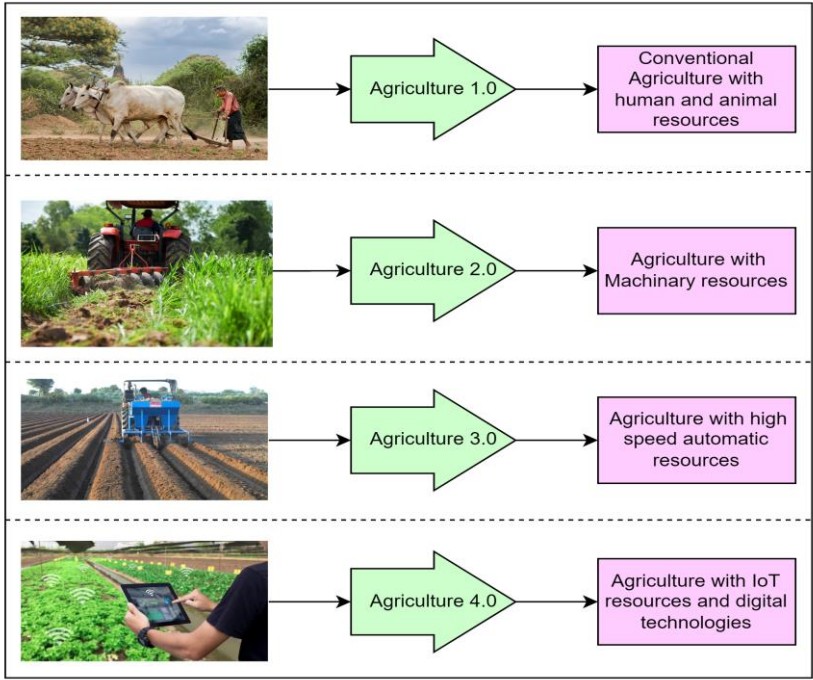

**Figure 1.** The uprising of Agriculture 1.0 to agriculture 4.0.

Figure 1 is used to explain the features of agriculture development to readers (from Agriculture 1.0 to Agriculture 4.0).

Agriculture 1.0: During the conventional agricultural age (1784–1870), which was dominated by animal and human resources, the fundamental problem with agriculture was its lack of operational efficiency.

Agriculture 2.0: During the automated agricultural period of the 20th century, resource waste was the major problem.

Agriculture 3.0: During the period of rapid advancement of automated agriculture (1992–2017), the key problem was the lack of intelligence.

Agriculture 4.0: the age of smart agriculture (which began in 2017 and is defined by unmanned operations) is primarily characterized by the application of contemporary information technology to both support and intelligently improve agriculture.

The term "smart agriculture" [15] refers to a new approach to agriculture that personalized service through the integration of contemporary information technologies such as the internet, big data, IoT, and many more [16]. In a nutshell, the new mode is an intelligent agricultural solution that fuses current information technology with agriculture. Although recent information technology opens up new possibilities for the progress of agricultural output, it also places heavy demands on security and privacy in the context of smart agriculture and poses significant obstacles therein [17].

More than a movement, agriculture 4.0 represents the next development in the sector's transition to a more intelligent, effective, and environmentally responsible one. The supply chain's daily operations produce a tremendous amount of data [18]. Most of the time, this information went unused, but thanks to big data and new technology, it can now be used to boost any crop's performance and productivity.

## 2.1. Advantages of Agriculture 4.0

The effects of smart agriculture are covered in the following sections.

Production volume: The use of smart technology in agriculture can significantly boost the number of products produced on the farm. This will contribute to feeding a growing population.

Production quality: The wellness and nutrition of people from all socioeconomic strata in the nation can be significantly impacted by the quality of food produced. A country's population will live longer and in better health with better food, which will improve its citizens' economic contribution.

Effectiveness of farming practices and resource use: The application of smart technologies to conventional agricultural operations can increase their effectiveness. In turn, this encourages more efficient use of agricultural resources.

Ideal cost of cultivation: It is achieved when there is a balance between quantity, quality, and efficiency in the procedures used. This results in a higher price for the agricultural output that is produced.

Reducing wastage: The farming industry, one of the main economic sectors, is in great part to blame for the massive quantities of food and other secondary resources that are wasted. Smart technology could be employed to track and cut down on this waste.

Ecologically sustainable: The reduced agricultural waste and improved agricultural process efficiency directly reduce the environmental and ecological footprint.

Effective use of Time: Timely delivery of the necessary insecticides, fertilizers, and other chemicals can provide timely and high-quality agricultural output with fewer losses thanks to smart agriculture.

The farming industry may be at threat by the following while deploying agricultural 4.0 technology:

- Theft of business and customer data.
- Taking resources under the control of sensors and gadgets.
- Destroying the objects that devices control.
- Damage to reputation if a data breach is disclosed.

Agriculture 4.0 may be at risk from infrastructure damage, sensor failures may affect poultry and cattle breeding, and control system hacks may affect greenhouse farming. All

of these could cause problems or malfunction in agricultural operations by harming the IoT architecture's hardware and software. In addition, data-collecting technologies face challenges from malicious assaults, unauthorized access, privacy breaches, and other issues.

The landscape has shifted, with AI and ML research concentrating on agro-based contexts, water management, livestock, and farmlands. Monitoring, control, and decision-making alternatives in the irrigation field repeatedly tried to save water and enhance productivity [8–13]. Some research concentrated on plant leaf disease [14], horticulture [15], vineyards [16], hydroponic [17], alert facilities [18], integration of IoT technological innovations [19], tracking resources [20], and cloud control [21].

### 2.2. IoT in Agriculture 4.0

Table 1 lists several well-known uses of IoT integration and utilization in agriculture.

**Table 1.** Use of IoT in agriculture and corresponding studies.

| Area | Years | Studies |
| :---: | :---: | :---: |
| Intelligent soil cultivation system | (2018–2023) | [22–26] |
| Efficient irrigation mechanisms | (2019–2023) | [27–30] |
| Smart fertilizer systems | (2018–2023) | [31–33] |
| Intelligent pest detection and treatment systems | (2018–2021) | [34–36] |
| Intelligent livestock agriculture | (2021–2022) | [37–40] |
| Smart harvesting system | (2019–2020) | [41–44] |
| Smart farm management system | (2018–2021) | [45–47] |
| Intelligent groundwater quality management system | (2018–2022) | [48–52] |

Intelligent soil cultivation system: This system would plow, weed, prepare the seedbed, and sow the field soil in order to prepare it for harvest.

Efficient irrigation mechanisms: This technology would automate the controlled artificial supply of water needed for plant growth.

Smart fertilizer systems: These automate the application of fertilizer to a field while allowing for precise control over the kind, amount, and timing of fertilizer.

Intelligent pest detection and treatment systems: These systems keep an eye out for pest infestations, analyze agricultural damage, and incorporate methods to manage the infestation.

Intelligent livestock agriculture: This involves employing cutting-edge technologies to breed animals and uses precision agriculture to enhance the quantity and quality of the produce.

Smart harvesting system: This system harvests a field effectively by using IoT-based methods.

Smart farm management system: This kind of technology would aim to offer analytics on data to increase field productivity and yield.

Intelligent groundwater quality management system: The final product is greatly influenced by the quantity and quality of the groundwater. As a result, this system uses IoT approaches to maintain appropriate groundwater levels.

Though there are many smart agriculture alternatives, they are still underdeveloped and offer a low degree of intellect. Many initiatives are limited in their automated systems, with the internet of things sending data to the base station [53]. There is also no implementation with the Internet in several cases, except that in a few cases, data is stored in the cloud by the local systems.

The multi-layer architecture of IoT used in agriculture 4.0 consists of four layers, which are the physical layer, network layer, edge layer, and application layer. Table 2 below gives a detailed description of Resources used in the multi-layered architecture of IoT in Agriculture 4.0 [54].

**Table 2.** Resources used in the multi-layered architecture of IoT in Agriculture 4.0.

| Layers in Agriculture 4.0 | Recourses Used in Layers | Description of Layers |
|---|---|---|
| Physical layer | Sensors and Cameras | Collecting data from the environment |
| | Actuators | Changing the state of the environment |
| | RIFD | Storing the data |
| | GPS | Tracking the location of the machinery |
| Network Layer | Connecting resources such as Routers | Connecting remote devices to transform data |
| Edge Layer | Security, interfaces, gateway | Uses Security protocols for ensuring data integrity, confidentiality, etc., gateways connecting devices with the cloud to store a small amount of data |
| Application Layer | Database, End users, Web tools, etc. | Storing data, exchanging information between the applications, and providing data accessing to the end users. |

The RFID, cameras, actuators, GPS, and all these devices are all implemented in the physical layer, which collects data from the farming field and digitizes them using an Analog to Digital converter (ADC). As the equipment at the physical layer does not compute and store the data, this needs to connect with the edge layer through the network layer. The edge layer includes various resources such as decision-making, gateway, data filters, data security, and interfaces [55].

The advancement of devices and technological developments will allow for the incorporation of more computer capabilities into systems. This type of integration aims to meet the various needs of agricultural mechanization, farming techniques, and sustainable agriculture [56,57]. A further issue is data security. This is very important to focus on data privacy, the trustworthiness of data, and the correctness of data from generation to decision-making.

*2.3. Research Motivation*

Three factors serve as inspiration for this article:

1. Smart agriculture is a new paradigm that integrates information technology with conventional farming as a result of the low productivity of traditional agriculture and the extensive usage of information technology. It has the potential to become the next big thing in agricultural development. Consequently, it is crucial to outline the current manufacturing model and particular studies [58].
2. Despite substantial research on smart agriculture, less has been conducted in comparison to industrial security solutions to analyze security problems.
3. It is crucial to examine the features of security concerns in relation to situations involving smart agriculture [59]. This article attempts to give a review of the security challenges raised by smart agriculture in light of the aforementioned variables, which inevitably results in a significant number of open research questions.

*2.4. Research Contributions*

The research contributions of the paper are as follows:

- We go over the advantages of using IoT in the farming sector and outline some of the potential uses.
- We provide a layered approach for smart agriculture that can be applied to any precision agriculture application.
- We suggest an agricultural sensor data management system that can gather, analyze, visualize, and manage sensing data in real-time.

- We present a blockchain-based authenticity monitoring technique to prevent erroneous control and information delivery.
- To improve network management, we proposed a simulated switch that supports SDN technologies.
- We present experimental results with different case studies from an open-source IoT platform integrating Ethereum blockchain and SDN technologies, demonstrating the efficacy of the suggested security architecture.

*2.5. Paper Outline*

In addition to the introduction, this paper mentions the overview of Agriculture 4.0 in Section 2. Section 3 provides important security risks from a multi-layer perspective. Section 4 describes the current state of security research in Agriculture 4.0. Section 5 presents the current scenario of security challenges in Agriculture 4.0. Section 6 describes the proposed agricultural security framework. The experimental setup configuration and architecture along with the experimental results presented in Section 7. The use of 5G technology in the scenario of smart agriculture is described in Section 8. Finally, Section 8 concludes with future challenges in Agriculture 4.0.

*2.6. Analytical Distribution of Referred Articles*

The referenced articles are compared in Figure 2 based on the place of publication. The distribution of cited articles by publication type is shown in Figure 2. Out of the total number of referred papers, 88 are original research articles from reputable journals, while 25 are from conferences.

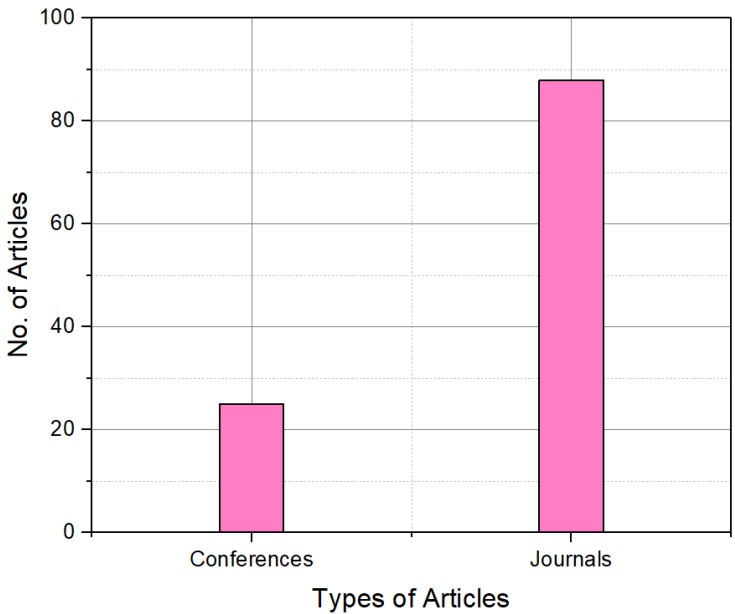

**Figure 2.** Types of Article distribution.

The frequency of papers discussing the agriculture 4.0 system is depicted in Figure 3. Due to our interest in current technologies, we have assessed 20 papers that were published in 2018, out of which five articles are based on agricultural security issues. In contrast to the 19 research papers on the agriculture sector and security aspects from the year 2020 that we have cited, we have cited 18 publications from the year 2019. With the papers becoming publicly available from various reputable journals for reference, we studied 46 modern technology-based quality papers in smart agriculture along with articles on security challenges in 2021 and 2022. In order to be updated regarding the current tools and technologies in this era, we also referred few articles to be published in 2023.

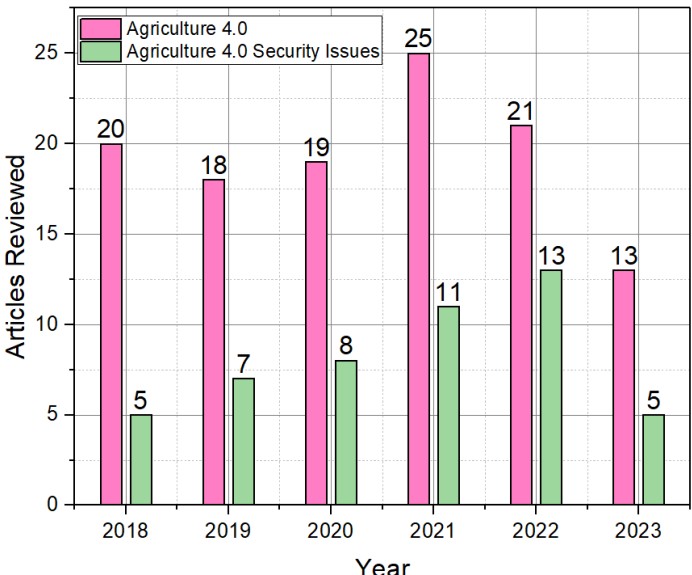

**Figure 3.** The total number of Related Research articles studied on Agriculture 4.0 and security challenges in Agriculture 4.0 from 2018 to 2023.

## 3. Agriculture 4.0 Security Threats in Multi-Layered Paradigm

There are several security issues in agriculture 4.0, starting from Storing data, processing data, and transmitting it through Internet connectivity. Figure 4 shows security risks in the intelligent agricultural system in a multi-layered paradigm. Security issues can be either unintentional or sometimes known. Animals, farm laborers, and machine tools can all easily gain access to agricultural environments and cause problems [55]. Most of the security threats are very common, but there some are unique to others who function in harsh-field environments, such as Intelligent Agriculture.

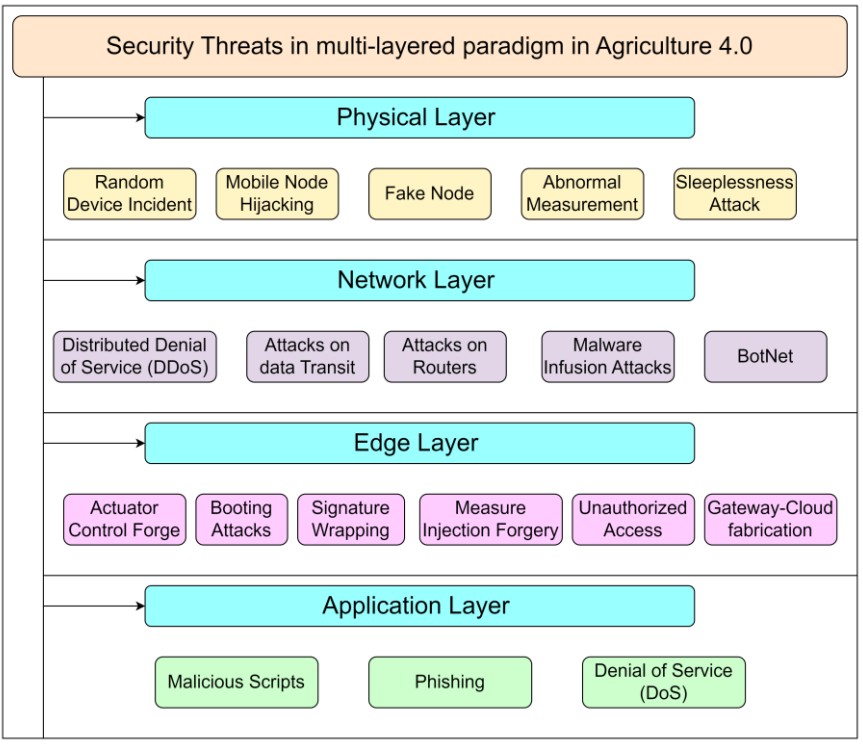

**Figure 4.** Security Risks at the multi-layered paradigm of IoT in agriculture 4.0.

### 3.1. Security Risks at the Physical Layer

This is primarily concerned with physical equipment such as sensors and actuators. Physical devices can fail due to unintentional or intentional human behavior, viruses, spyware, or cyber-attacks. Smart farming applications use a wide range of sensors and technologies, and this diversity opens up several security risks, including the following:

Random device incidents: This is the unintended alteration of physical equipment that causes it to deviate from its normal performance. The physical equipment may be damaged by farm machinery, causing short- or long-term physical damage, as a result, generates false data and affects data accessibility.

Mobile node hijacking: It is the theft of farm equipment such as tractors, drones, and sowing robots. Drones can indeed be pesticides, fertilizers sprinkling could be performed by Drone, and robots could do weeding detecting disease. The intruder may access and use the farm equipment remotely without having access permission.

Fake node: An intruder introduces imitation or harmful nodes into the automated agriculture field to disturb the smooth functioning of the system [55]. This attack could be triggered by capturing a node and making a replica of it. This type of attack typically aims to either alter or modify data-shuttered devices and applications.

Abnormal Measurement: It senses unusual observations due to data falsification, energy exhaustion, radio radiation, detection of various connections, extreme storms, malfunctions, or wrong inputs. Incorrect data can jeopardize making decisions that result in the analysis of data will be inaccurate, which minimizes system correctness.

Sleeplessness attack: The sleeplessness attack sends a series of seemingly legitimate requests in order to keep the devices awake for prolonged life. As a result, the device's battery runs out, and the device stops working [56,57]. When the devices are turned off, the observed information is not properly sent, jeopardizing decision-making and system efficiency.

### 3.2. Security Risks at Network Layer

The objective of this layer is to send the generated data by the sensors from the physical layer to the most trustable computational unit, which is the cloud. The most common attacks, which affect the resources used in the network layer, are focused on [58,59]. The security risks in network layers are:

Distributed Denial of Service (DDoS): It attempts to affect access to service providers by overburdening the communication link or manipulating protocol flaws, which cause resources, such as CPU and memory collapse [60].

Due to resource exhaustion brought on by a DDoS attack, the victim server is forced to block connections from new reliable clients. The victim server's buffer capacity or bandwidth may have reached its limit of resources. The overall probability of resource scarcity at the target end is provided by Equation (1) [15].

$$P_{Attack} = 1 - (1 - P_{Bandwidth})(1 - P_{memory}) \tag{1}$$

where,

$P_{Attack}$: Probability of total attack,

$P_{Bandwidth}$: Probability of Consumption of bandwidth

$P_{memory}$: Probability of Consumption of memory

Attacks on Data Transit: These types of attacks aim to retrieve information shared between network devices in order to obtain confidential information [61]. An adversary could intercept traffic via malicious routers or man-in-the-middle attacks [62]. Interception of traffic reveals confidential data such as private keys, access credentials, or digital signatures.

Attacks on Routers: They plan to change network routes in order to gain traffic control. Inimical nodes in IoT networks might attempt to reroute communicating paths during the

process of transmission. As a result of attacks, the receiver might receive information lately, partially, or incorrectly, or not at all [63,64].

Malware Injection Attacks: The malware injection attack [34], in which malware is injected by an attacker into a connected computing device. This is a very common attack on smart farming. Malware is a very significant threat in distributed projects because it automatically spreads through the system, making it an appealing target for intruders. Malware damage comes in multiple varieties of sizes. Malware can steal data about farm material usage, buying information for fruits, vegetables, and farm animals, data about farm equipment, and so on. It could hire automated devices to form a botnet that would be used to commit harmful attacks under the influence of an intruder.

Botnet: At each architectural layer of the precision farming system, there are numerous IoT devices. These machines are vulnerable to attack and could be taken over by a centralized malicious system. This results in a 'Botnet of Things' (BoT) [35]. A mutant farm workforce IoT equipment that is already infected [36] is easily used to infect other networks via various communication links, and thus automated farming could become an internet of destructibility for cyber illegitimates. The security features are not added in many of the Precision agriculture devices, but still, they are used frequently for establishing adequate data security defensive system frameworks.

### 3.3. Security Risks at Edge Layer

The edge contains important aspects that monitor, and control modules communicate with all layers and access resources. The physical layer's large volumes of data can be processed locally rather than centralized in the cloud. The following are major edge security issues.

Actuator control Forge: It introduces false measures of data into the system in order to manipulate it. Typically, the data is received by the gateway in plain text. Corrupted data will lead to poor decision-making. An opponent, for example, could inject incorrect soil moisture measurements into a smart irrigation system to manipulate it.

Booting Attacks: Devices are vulnerable to attacks due to a lack of security processes during boot [37]. Memory cards and flash drive sticks, for example, may detect dangerous codes which run at startup [38]. Suspicious booting practices could launch numerous attacks on the edge with insufficient protection.

Signature wrapping: The attacker modulates the cipher text by inserting a bogus element that allows you to perform a random Web Application requisition while identifying as an authenticated user [39].

Measure infusion Forgery: It is the injection of wrong specifications into a system in order to modify it. An attacker sends known data patterns through actuators to the cloud storage, which is then injected into the system through computational units. Corrupted data could lead to poor decision-making.

Unauthorized access: The process of validating is a critical security component. The authentication process is a technology that ensures security and privacy, honesty, and accessibility [40]. Several agricultural practices, however, use gateways with insufficient or no access controls.

Gateway-cloud fabrication: The cloud and the gateway are linked by ISPs or wireless connections. A network attacker could fabricate cloud requests by interrogating a gateway. The attacker could use these requests to change parameters in precision agriculture, control queries for sensitive services, or interpret system equipment. The cloud can implement security services as it has computational power. In order to ensure maximum system reliability, these strategies must be incorporated into the system.

### 3.4. Security Risks at Application Layer

This layer provides end-user assistance and data to process and make system decisions. This layer's security concerns are application-specific, focusing on preventing data theft

and ensuring privacy. Some of the attacks that might affect cloud services and applications are listed below in terms of security.

Malicious scripts: Malicious scripts can deceive clients, infuse malicious data, access confidential material, and compromise security features. This type of attack is frequently carried out by cybercriminals for specific, economic, or political reasons. They can damage or destabilize operational processes by using malicious scripts, presenting unnecessary ad campaigns, and stealing money [41].

Phishing: It is an attack that attempts to obtain highly classified personal information, such as an Identity and password, fraudulently. Phishing typically targets end-users through fraud and sites [42]. The attacker also affects decision-making and internal operations in many cases. The most effective protection would be for users to remain vigilant while surfing the web [43].

Denial of Service (DoS): This threat disrupts service by overburdening traffic on the network or flooding the provider with repeated attempts [44]. Because of flaws in security setup, an attacker can launch this attack from the Web or a sub-system.

## 4. Security Threats in Modern Agriculture

Farmers typically carry out the developed smart system initiatives in physical fields or green fields. In Agriculture 4.0, most efforts are focused on water processes, disease monitoring, seed production, and record keeping. It can be either automatically controlled or manually controlled. In both cases, the system monitors the environment with sensors and changes it with actuators. Some initiatives only digitize farms, whereas others incorporate industry 4.0 or IoT technologies. As an immature industry, open-field agriculture is the focus of this work, with security issues restricted to access control and web-encrypted communications.

It is too difficult to execute different security measures in rural areas since there is a shortage of modern technical infrastructure and advanced technology. Many difficulties can arise if security measures are attempted in rural agriculture. Table 3 lists a few of the security challenges that are prevalent in rural locations.

**Table 3.** Rural-Agriculture Security Issues.

|  | Rural-Agriculture | Security Issues for Rural |
|---|---|---|
| Resident | security awareness is less | Hard for Facing security risks |
| Facility | Cost and energy consumption are less | Hard for imposing Strong security mechanisms. |
| Production data | Difficult to measure | Identifying unauthorized access is hard |
| Production cycle | Depends on the growth of the crop | Face heavy loss because of less security |
| Transport | Farmland environment, bad traffic | Delayed availability |
| Management system | Weak security due to weak infrastructure | hard to control attackers |
| Communication | Base stations are less | Hard to identify base station attacks |

Recently most of the agriculture 4.0 projects have implemented IoT, but the security features in that have become disabled by default. It has been introduced that a server controls and manages these devices remotely via a web application. Opportunistic opponents did not know details about security resources. They gain unauthorized access to the system, insert fabricated measures, fabricate actuator controls, or report earlier attacks to divert the system from its normal functioning [45–64]. The author proposed a smart watering system that remotely controls and observes the flow of water in the farm but is not focused on security risks to detect malicious attacks and unallowed access that manipulate the monitored data. To monitor the growing environment, ref. [65] introduce a Hydroponic Farming Ecosystem (HFE). It is automatically controlled, and the client can monitor the farming via a web interface. These automated systems require high-security concerns to

detect or avoid random sensor incidents, weak sensors, wrong data injection, and other attacks, which could corrupt data and affect system reliability.

Similarly, in [66], the author focused on creating a system for monitoring fields based on humidity, soil moisture, light levels, and temperature. Irrigation can be controlled manually or automatically via web or mobile systems. An intelligent solution for detecting diseases in leaves focused in [67]. By using sensor data and camera images, the system identifies leaf diseases. A mobile or web application is used by the end-user to interact with the system. This paper does not go into detail about the implementation of security.

Similarly, ref. [68] proposed an intelligent agricultural system for information interchange among farm equipment. The article discusses the system's construction but makes no mention of manual intervention or remote control, nor does it concern any security challenges. As this is a data exchange system, the DoS attack can affect the network layer. Ref. [69] provides NETPIE, a solution that gives agricultural product information. The growing environment can be controlled and monitored with a set of physical devices. The generated data is analyzed saved in a QR code, and which is then available to the customer. NETPIE does not discuss security resources. Ref. [51] Introduced monitoring of crop fields and watering using IoT, which measures data and controls data to avoid unauthorized access to these data but could not protect the other devices, which leads to weak security, which might result in a major loss. Ref. [52] Proposed a system known as SEnviro, which observes the crop field and detects diseases remotely. This system does not focus on the prediction of disease, but unauthorized access is controlled here.

According to the relevant literature, from a security standpoint, farming equipment is used without security features. Furthermore, the gateway contains no security information. Transmission privacy or device authentication is not presented in any of the above-discussed systems. Access control and management and encryption of data need to add security technologies to Information dissemination. Smart agriculture is currently a convenient target for cyber criminals' agents. Attacks can be motivated by a variety of factors, including an advertisement, intellectual, or even terrorist reasons. Terrorist groups, for example, can cause ecological loss to a country, economic chancers may take part in market manipulation, and workers can attack a range of factors [53].

### 4.1. Data and Device Security Issues and Threats in Agriculture 4.0

As was previously mentioned, harsh environmental factors that can seriously harm electromechanical equipment, such as extreme heat, moisture, moisture, shocks, and other anomalies, have a direct impact on the agricultural sector [70]. The majority of these sensors are prone to failure, making it feasible for them to produce inaccurate data and instructions that could cause a manufacturing disaster [71]. Additionally, sensors and network equipment are predominance reachable. This poses a serious risk to farmlands since anybody with bad intentions can access them and corrupt or harm them to cause them to malfunction [72].

Another potential security concern is the online transfer of the information gathered by IoT sensors and other agricultural machines. Major security concern in the agriculture industry is security, privacy, and control [73], as farmers may incur a significant financial and personal loss in the event of a data breach. Ref. [74] Divides the threats to confidentiality into four distinct categories:

- Intentional data theft through smart platforms and applications that do not adhere to security and privacy standards;
- Internal information thefts from a stakeholder in the supply chain intended to harm an agri-business or a farmer;
- Unethical data sales intended to reduce profits for farmers or to harm them; and other threats.
- Unsupervised foreign access to private and sensitive information using drones, sensors, and cameras with the intention of utilizing them to harm farmers or jeopardize public safety [64,75].

Every application of modern technology, IoT in agriculture, carries the aforementioned dangers, weaknesses, and threats. A more thorough and complex explanation of the risks and their sources in aspects of cyberattacks in the agro sector is provided in the following paragraph, which is primarily focused on illegal behavior and exploited vulnerabilities in ICT systems.

### 4.2. Cybercrime and Cybersecurity in Agriculture

A cyber-attack on an agribusiness or food industry is increasingly likely today thanks to automation and the widespread use of internet-connected equipment, which give potential (cyber) criminals greater chances in fields that were previously too remote or difficult to physically access [61]. Consequently, a potential cyber-attack or security problem in an agricultural organization could have serious human or financial repercussions. This implies that pertinent security concerns brought about by the widespread and ongoing usage of IoT, as well as the emergence of agro-terrorism, require adequate attention, better equipment, and solutions [62].

The bulk of system operations in the agriculture sector are network-based, and frequently these systems are not protected from cyber threats, which can have substantial financial and security ramifications [76–86]. Five important elements, as identified in [87], strengthen these risks in the agricultural sector:

- Increasing the farm consolidation highly depends on technology;
- The joint ventures of the food supply chains, allow manufacturers to conduct processes and trade products directly;
- Food-related technologies in intelligent markets depend on more components, which increases their vulnerability to errors and malfunctions.
- Effective monitoring of food-related systems, social networks, and industries in a safe, dynamic, and almost real-time manner is lacking, making it difficult to identify serious digital and security vulnerabilities that could be the root of significant data breaches and system defects.

Due to the heightened risk of cybercrime, the agriculture sector may be impacted by such threats in a number of ways. Categorize these methods as in [84]. Threats of tampering include delivery disruption, confidential information interception, formula modification, and delivery disruption. These risks are undoubtedly closely related to the adoption and application of technological advancements in the agriculture and food sectors.

Cooper et al. and Bogaardt et al. identify the following as the primary technical advancement areas that are commonly used in the agri-sector and contain vulnerable places for harmful cyber-attacks [86–88]:

- Radio Frequency Identification (RFID); wireless communications (such as Wi-Fi);
- Sensors for the infrastructure, soil, and crops;
- Drones and other unmanned aerial vehicles (UAVs);
- PA-specific automation solutions (such Real Time Kinematic Technology);
- Portable electronics (such as laptops, smartphones, and GPS trackers);
- Smart agriculture and vertical agriculture;
- The use of AI in conjunction with biotech and nanotech (AI).

Table 4 provides a summary of the study performed on different attacks in agricultural 4.0 due to cyber security concerns.

**Table 4.** Summary of the study performed on attacks in agricultural 4.0.

| Attacks | Agriculture Consequences | Years | Studies |
|---|---|---|---|
| Physical Attack Replay Attack Masquerade Attack | The collecting of data on the kind and potential applications of equipment for agricultural projects violates privacy. | (2019–2022) | [62,76–81] |
| Dictionary attack, Session Hijacking, Spoofing | Attackers with forged identities who can pass as legitimate or authorized people can access the precision agriculture system. | (2019–2023) | [62,79,82–84] |
| Malicious Code Attack Repudiation Attack | A situation in which services, authentication mechanisms, or data transmissions are refused through the system's nodes may result from the repudiation of information, which enables an intruder to repudiate all the energy usage, information-generating, and manufacturing processes of an agricultural production ICT system. | (2018–2021) | [62,79,85,86] |
| Tracing Attack Brute Force Attack Known-Key Attack | Because of the confidentiality breach, unauthorized access to crucial data could result in the theft of significant data and pose serious risks to the privacy of the users of the involved agriculture system. | (2020–2023) | [29,62,79,87,88] |
| Forgery Attack Man-In-The-Middle Attack (MITM) Trojan Horse Attack | Information about agriculture technology or smart farming techniques may no longer be accurate or dependable due to potential unlawful or improper changes in the reliability of data or resources. | (2019–2023) | [62,89–96] |
| Denial of Service (DoS) attacks (SYN Flood, Ping of Death, Botnets) | Attackers have the ability to halt the functioning of the established smart farming network or even set up services that are inaccessible to farmers. | (2019–2022) | [62,79,92,97,98] |

### 4.3. IoT Vulnerabilities, Risks, and Threats in Agriculture

IoT is made up of four main systems that make it possible to communicate between nodes. Sensing technology, IoT gateways, cloud servers, data storage, and mobile applications for remote control make up an IoT system. The architectures of IoT systems in agriculture come with current and new threats and vulnerabilities. These flaws can be linked to IoT device hardware and software problems, communication protocols, and also data processing and storage solutions (found, for example, in cloud infrastructures, data centers, and smartphones) [2,11].

According to [99–101], the following is a concise but accurate classification of the primary causes of low security in the IoT:

- Firmware that is not patched and/or default passwords that have been used for a long time allow for device compromise in an IoT network.
- Because of the limited computing power of smart devices and vendors' efforts to keep their prices low in a cutthroat market, it is difficult to incorporate complicated cryptographic algorithms.
- Flaws in the routing protocols used by smart devices (such as Bluetooth and ZigBee);
- The Wi-Fi Protected Access (WPA) protocol's outdated, low-security version, which is frequently still in use;
- Conducting passive vulnerability detection using search engines;
- The risk of assembling millions of smart devices into a potent botnet (such as Mirai), given how simple it is to find vulnerabilities via internet scanning;
- A general disregard for the security of smart gadgets

Finally, a variety of criteria can be used to categorize IoT vulnerabilities, hazards, and threats. These risks apply to all IoT applications, including those in the agricultural sector. First, internal vs. external risks and passive vs. active risks are the two basic classification groups of security concerns in IoT systems in the agriculture sector [79].

IoT hazards are categorized in another intriguing way using a layer-based system. According to [22], each of the levels below integrates many technologies that could lead to a variety of security threats, vulnerabilities, and/or breaches. The application layer, middleware layer, internet layer, access gateway layer, and edge technology layer are the five main layers that make up the IoT architecture.

## 5. Existing Research on Security in Agriculture 4.0

In [54], the author states that The United States Department of Homeland Security issued a report emphasizing the significance of agriculture 4.0 and the associated internet security threat and possible attacks.

The report emphasizes the information security model of confidential information, credibility, and accessibility in farming. The researchers of [55] suggested a new access control solution that combines session keys and public keys to speed up the encrypted communications tasks. It yields quick and lightweight appropriate control measures that are ideal for precision agriculture communication. Ref. [56] Proposed a framework for understanding vulnerabilities in emerging technologies and their application in an intelligent farming environment. The goal of the approach is to determine the level to which intelligent farming emerging innovations are prone to cyber. It assesses the threat prediction model using the common vulnerability scoring system (CVSS).

Nowadays, The Blockchain is not only used in cryptocurrency but also utilized in other fields and different applications such as in the medical sector, agriculture sector, etc. [57–59]. This article concentrated on the application of blockchain technology to food safety. For tracking and monitoring the food items and stages of production, such as cultivation/breeding, raw resources, processing, transporting, warehousing, and selling, the author developed a system [60]. In addition, the system employs a variety of sensor-based equipment to adopt sustainable recording and confirmation with sensor-based verification. Ref. [61] Proposed a blockchain-based easy and simple security framework for smart Greenfield farming. Each IoT node can nominate the cluster member's leader, which helps to avoid a hot spot problem if an intruder focuses on the leader node. Furthermore, the authors discussed the security risks of their suggested scheme, which adhered to the traditional confidentiality, integrity, and availability principles. Ref. [62] proposed a cost-effective IoT-based security monitoring system. The physical layer is the primary concern of this system of precision agriculture, where data are collected from sensors. These data are sent to a controller, which analyses it to take action, such as activating the actuators for water sprinklers on farms. Table 5 describes some of the recent research which has already been performed on security issues related to smart farming.

Gupta, M et al. explained [102] the necessity of clouds applied in vehicles in this time- and location-sensitive context and present an extended access control oriented (E-ACO) architecture that is pertinent to IoV. Here, they describe using scenarios to highlight access control needs in our vision of cloud-assisted connected automobiles and vehicular IoT, as well as approaches to various access control models that may be applied at different tiers of the E-ACO architecture and in the authorization framework.

Gupta, M et al. [103], in their work, describe an edge infrastructure-based solution to V2V and V2I communication that is safe and trustworthy in place of direct peer-to-peer connection. To approve, check, and verify the authenticity, integrity, and anonymity of communications sent in the system, they add trusted cloudlets in this section. Along with a proof of concept implementation of the suggested approach on the AWS IoT platform, they also describe a formal attribute-based model for V2V and V2I communication, known as AB-ITS. This cloudlet-supported architecture supports crucial use cases such as an accident or ice-threat warning and other safety applications. It supplements direct V2V or V2I communication.

Sontowski, S. et al. [104] presented a DoS attack that can impair a smart farm's operation by interfering with the functioning of installed on-field sensors. They specifically covered a Wi-Fi de-authentication attack that makes use of flaws in IEEE 802.11, where

the management packets are not encrypted. A development board for the ESP8266 maker focus When a Raspberry Pi is attached, WiFi Deauther Monster is used to disconnect it from the network and stop sensor data from being transferred to a distant cloud.

The notion of Activity-Centric Access Control (ACAC) for smart and connected ecosystems was first proposed by Gupta, M. et al. in [105] and discussed in this article as a first step. Here, they examined the idea of activity with regard to cooperative and dispersed yet interconnected systems, identifying the many parties involved as well as the crucial elements to consider when making an activity control choice. They provide an initial framework for developing activity control expressions that may be used by various smart objects in the system.

The biggest obstacle to the growth of cloud computing is security worries. Information security continues to have an impact on the market even though it raises inconveniences related to information security. The threat of security breaches in the cloud infrastructure must be understood by clients. Cloud-computing capacity, networking, and computing powers are brought to the edge by fog computing. One of the most critical challenges with fog computing platforms is safety and security. Lightweight devices have benefited greatly from edge computing's ability to complete difficult tasks quickly, but the technology's hasty development has led to a general disregard for safety risks in edge computing stages and the applications they allow.

In our proposed framework we have focused on the threats in fog computing-based IoT environment via blockchain for the application in agriculture 4.0.

**Table 5.** Related Research performed on Security issues in Agriculture 4.0.

| References | Years | Scenario | Technology & Security |
|:---:|:---:|:---:|:---:|
| [73,75,92] | (2022–2023) | Smart equipment, Smart irrigation | Automation, IoT<br>Theft detection by motion detector |
| [84,97] | (2019, 2021) | Efficient and secure cluster routing for IoT-based smart agriculture applications | WSNs, data encryption<br>A strong communication channel and symmetrical data security for security between farming devices |
| [35,95] | (2021, 2023) | Supply Chain Tracking, Aerial Crop Management, Livestock Safeness, and Maturity Tracking, and Irrigation | Smart agriculture IoT<br>Cyber security in smart agriculture |
| [76,87,93] | (2019–2021) | Security and privacy in green-IoT-based agriculture | Blockchain, IoT<br>Privacy-oriented blockchain-based solutions |
| [106] | (2021) | Ecosystem for smart agriculture | IoT<br>Authenticating IoT sensor device and sending encrypted data |
| [79,100] | (2021, 2023) | Intelligent and secure smart irrigation system | IoT, blockchain<br>Decentralized storage of irrigation and plants database by implementing the concept of blockchain |
| [48,96] | (2018, 2020) | Precision agriculture threat prediction model based on Common Vulnerability Scoring System | Parameters detection, IoT system<br>A prediction model framework for cyber-attacks in precision agriculture |
| [68,94] | (2021, 2020) | Precision agriculture-based multilayered security and privacy architecture | IoT, AI<br>A holistic study on security and privacy in a smart agriculture ecosystem |
| [59,91] | (2021, 2019) | Swarm robotic systems for precision farming | Blockchain<br>Public key cryptography by blockchain |

**Table 5.** *Cont.*

| References | Years | Scenario | Technology & Security |
|---|---|---|---|
| [63,98] | (2023, 2022) | An overview of the security requirements, problems, thread model, stack challenges, and attack taxonomy for smart agriculture | IoT Livestock management, precision farming, greenhouse monitoring |
| [104] | (2020) | DoS attack in a smart farm's operation by interfering with the functioning of installed on-field sensors. | IoT, on-field sensors and autonomous vehicles |
| Proposed Framework | Proposed | Agriculture 4.0-based multilayered security and privacy architecture. DDos Attack against blockchain Network | IoT, Fog Computing, Blockchain, DDoS mitigation programme on the SDN controller. |

## 6. Proposed Security Framework

The proposed security framework for agriculture 4.0 is described in this section. The block diagram of the proposed security framework is illustrated in Figure 5.

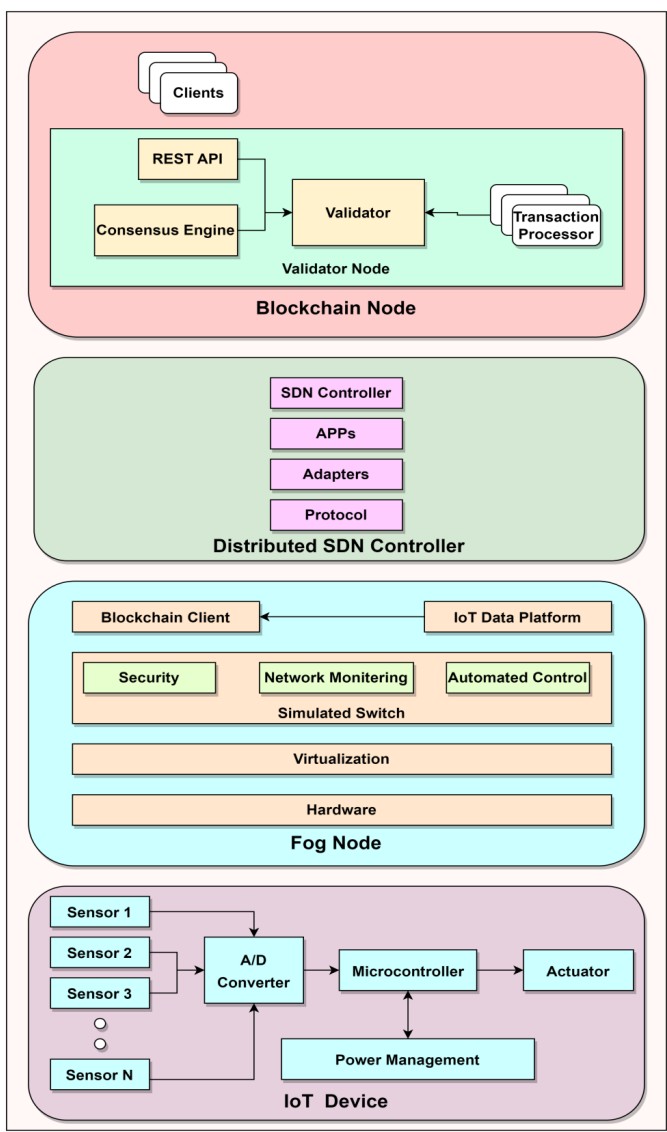

**Figure 5.** Proposed Security Framework.

### 6.1. Sensor Layer in Agriculture 4.0

This layer's major function is to acquire meaningful information from the external environment via sensors to take required action against it. This sensor layer has a variety of Sensor nodes of various forms and sizes. As a result, they all use the same basic components, such as input and output components, information handling units, communication segments, and many more units for the management of power sections. To analyze and store all detected data, it is sent to the relevant fog nodes. Furthermore, Sensor nodes receive instructions to undertake agricultural operations.

### 6.2. Fog Computing Layer

This layer is made up of multiple fog nodes, which are insubstantial equipment that carry out processing, communication, and storing tools to do operations that an end device cannot perform. Not every piece of data that is gathered becomes processed on the cloud. Rather, this layer hosts all authentic analysis and latency-sensitive applications.

Virtualization: Recognizing that configuring several devices, in the same manner, might be time-consuming, the nodes can be contained inside a Virtualized environment for a more pleasant setup for all subsequent fog nodes.

Agricultural Sensor data monitoring: This node is going to collect real-time sensing data, analyze it, visualize it, manage devices, and store some of it locally. The user can obtain information from the fog layer while smoothly synchronizing cloud-directed data.

Block-chain Client software: This consists of a piece of programs that is installed on the fog node. Its primary goal is to sync authentic sensitive Sensor data to the Blockchain. It is in charge of producing transactions, batching them, and sending them to the validator so that they can be aggregated into blocks and recorded to the Blockchain.

Simulated Switch: Additionally, this node is used as a transmitting device with the support of virtual switch software, which provides NFV and SDN technologies to enhance network services by establishing the routing features for computational extension and control. Some network virtual functions include security, quality of service (QoS), and automated control. The main objective is to create switching layers for virtual machine environments while continuing to support the OpenFlow protocol [107], which allows this node to interact directly with the SDN Controller; an outstanding demonstration of such implementation is Open vSwitch, an interactive multilayered software switch.

### 6.3. Distributed Network Using SDN

This network is made up of numerous geologically dispersed and SDN controllers linked to it that operates as a common Networking Operating System (NOS), combining nominally autonomous operations with centralized control [108]. The main component of an SDN network is the SDN controller. This can help with network monitoring by managing all transactions among network and application devices, allowing it to manage and adjust network flows more efficiently in response to new requirements [109]. It maintains a consistent glimpse of the network from afar, delivering data to fog nodes via one APIs and application via other APIs (MQTT, CoAP...) [109]. In addition, the SDN controller employs a Block-chain contractor to properly secure flow tables of SDN in the Block-chain, hence avoiding rule counterfeit.

### 6.4. Block-Chain Based Network

To keep all the nodes up to date, all Blockchain modules are interconnected and constantly exchange the most updated information in the Blockchain among themselves. Smart contracts will trigger and enforce relevant actions based on the data provided, allowing for verifiable transactions in the Blockchain when such circumstances are met without the requirement for third-party involvement [110,111]. The Ethereum Blockchain platform serves as the foundation for the blockchain network architecture because it provides a highly flexible and modular design that separates the runtime environment as per the

application domain, permitting agreements to define the business rules for the application that are not necessary for the core system's fundamental design.

The Blockchain complete node is made up of the following components:

1.  Node for Validation: This is a key component that verifies batches of transactions before combining them into blocks. It also ensures that candidate blocks are added to the Blockchain version for each node.
2.  Blockchain Software: it determines which transactions or operations are permitted on the block-chain and contains the following:

    -   Database model: it defines relevant operations and the description of the transaction's payload.
    -   Transaction Processor: defines the business logic for different applications, validates batches of transactions, and modifies the blockchain system based on the application's rules.
    -   Client: it specifies the service client functionality, which the client develops and delivers to the validator. The client also shows blockchain information.
    -   REST API: This protocol is used to communicate within the clients and the transactions processing system.

3.  Digital Signature: To ensure transaction integrity, the data used in the transactions is hashed using SHA-256 and then encrypted using the sender's secret key to produce a signature. The processor then validates the transaction by verifying this signature.

## 7. Case Studies and Experimental Results

To test the security framework, we allowed DDoS attacks against the Blockchain Network. A distributed denial-of-service (DDoS) attack targets websites and servers by disrupting network services. A DDoS attack attempts to exhaust an application's resources.

We provide an experimental setup configuration shown in Table 6, as well as the simulation architecture is shown in Figure 6.

**Table 6.** Experimental Setup Configuration.

|  | **HM** | **VM** |
|---|---|---|
| CPU | Intel Core i5-6300U @ 2.4 GHz | 2 vCores |
| RAM | 8 GB | 2 GB |
| OS | Ubuntu 18.04 × 86 64 | Ubuntu 18.04 × 86 64 |
| Virtualization | Docker Docker-compose | Docker-compose Docker |
| Applications | ThingsBoard Open vSwitch ONOS Mininet Hping sFlow | Ethereum Supply Chain |

Host Machine (HM): Here we have installed an open-source based IoT framework called ThingsBoard, which make it possible to construct fog layers while seamlessly syncing with the cloud, on the host system and constructed a scenario of a smart farm with two silos, each with a different number of IoT devices. Every second, each gadget reports a pseudo-random value. We utilized a code to simulate a Blockchain client, which accepts the provided values and generates transactions. Mininet network emulation tool is then used to commit the transaction onto the Blockchain via the Open vSwitch module. We also connected the ONOS SDN controller and sFlow4 collector to the Mininet virtual network.

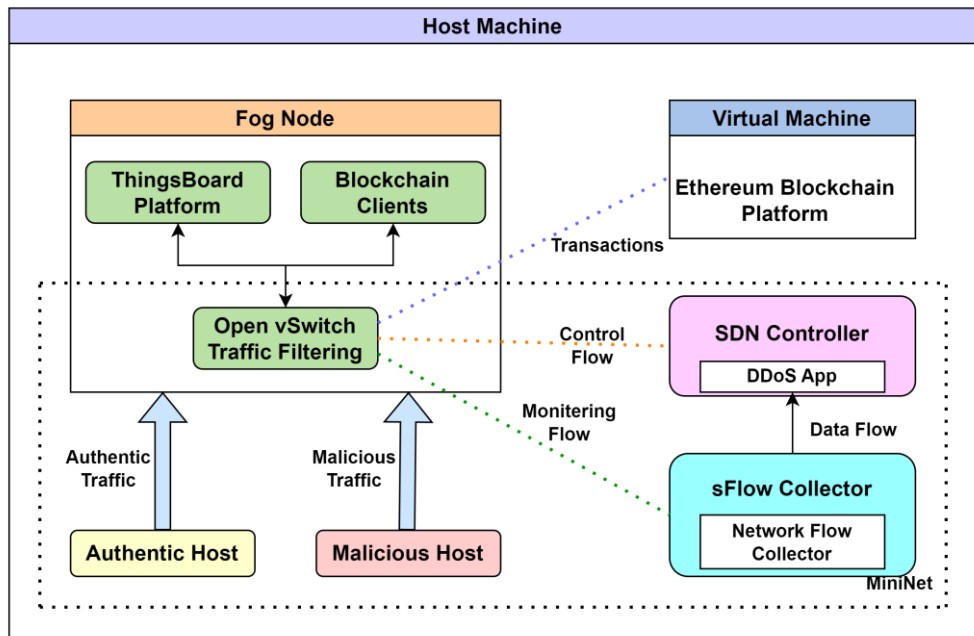

**Figure 6.** Experimental Setup Architecture.

Virtual Machine (VM): We have installed the Ethereum Blockchain network and its Supply Chain AssetTrack application in the Virtual Machine. To keep records of their updated information, we formed a user for the host system (fog node) and various assets. Every reported value will be used by the blockchain client to update the specified asset in the Ethereum blockchain.

To put our framework architecture to the test, we conducted three case studies.

Case 1: We examined the platform's usual workflow, which means we did not conduct any attacks on the Ethereum blockchain.

Case 2: Using the Hping programme on the host PC, we launched a DDoS assault on the Blockchain network. We implemented a DDoS mitigation programme on the SDN controller.

Case 3: We launched a DDoS attack against the Blockchain network, by turning off the DDoS mitigation programme in the SDN controller.

During the simulation, we noticed two crucial metrics in each case:

- The number of network packets received by the Blockchain per minute.
- The number of transactions published in the Blockchain network.

The results are shown in Figures 7 and 8, respectively. Taking the case 1 findings as ideal, with 120 published transactions in 10 min at an average of 100 Kbps; the third scenario revealed a really dismal performance with about 45 published transactions, representing a loss of more than 60% compared to the 1st case, and an unreliable Blockchain network. To understand the outcomes of the second scenario, we must first recognize what took place in the network during the test. The Open vSwitch sends all network traffic data to the sFlow forecast system, a real-time reporting system that gives real-time notice of network assaults, involving attackers and target information, to the SDN controller security framework via commands. The SDN controller interacts with the switch to discard DDoS traffic while allowing valid traffic to continue through. The attack is stopped fast enough to prevent it from rapidly growing and resulting in 95 blocks published, which is approximately 40% more than the 3rd case.

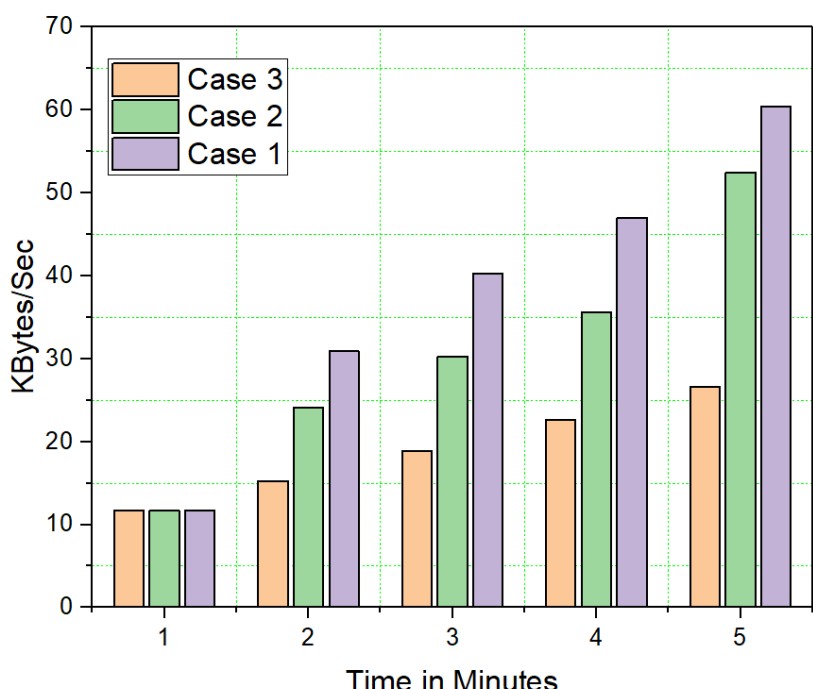

**Figure 7.** Number of packets received by Blockchain per minute.

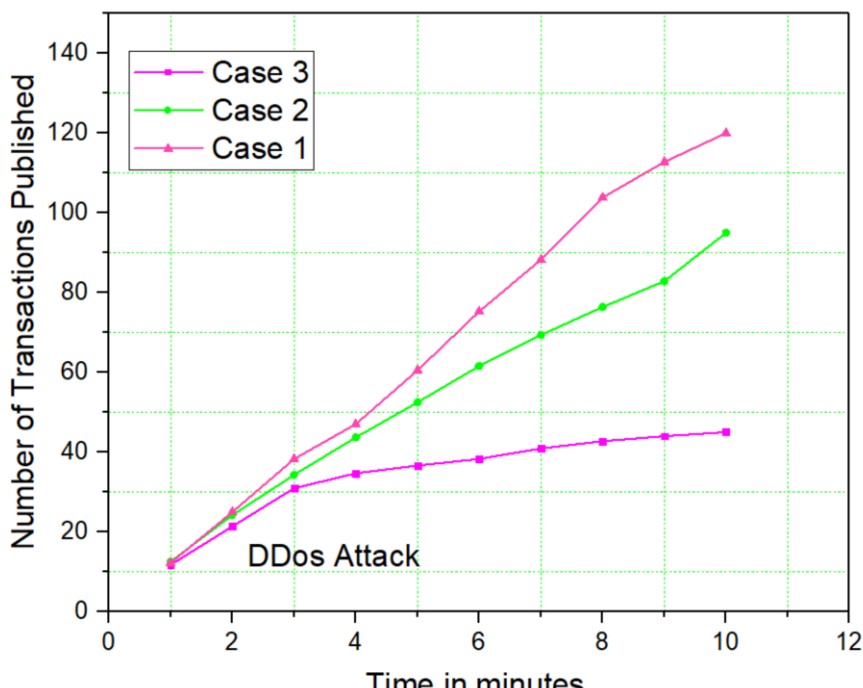

**Figure 8.** Number of transactions published in Blockchain per minute.

## 8. Smart Agriculture 5.0

With 100 times faster upload and download speeds than "fourth generation of mobile technology (4G)" and "fourth-generation long-term evolution (4GLTE)" technologies and latencies as low as 1 ms, the "5th generation mobile network (5G)" is a new technology for wireless that can connect a million gadgets per square kilometer. In order to exchange data effectively, the millions of smart gadgets that make up the precision agricultural networks must be connected to one another over a given piece of land.

Therefore, the Agriculture 4.0 scenario is particularly well adapted to 5G technology [112–114]. A framework for the digitization of agriculture networks and function modules for an intelligent irrigation system was provided by Meng and Cheng [115]. Tao and Donglin [116] also investigated how 5G technology may be used for agricultural planting, livestock farming, and traditional farming growth [117–121].

## 9. Conclusions

Modernization of agricultural methods is required to increase productivity levels while conserving nature. Precision agriculture could improve farming works by controlling actuators more efficiently, enhancing the use of resources, increasing revenues, and minimizing expenses. In order to accomplish this objective, IoT technologies have to incorporate more computing performance, such as edge computing, big data handling, automated resources, and safety features. As limited devices sense a huge amount of data and forward it to the gateway or the cloud, security requires special attention. The data must be secured throughout the agricultural system, from diagnosis to risks and inventory.

Even though numerous security issues may affect farming production, they still include a few security measures. This could be because these technologies were in their initial phases of development. For most of the points, only computing resources are incorporated, and that has limited computation power. As a result, security mechanisms are very rare in the current list of requirement specifications. Reaching a higher level of agricultural development, on the other hand, necessitates technologies with security features that provide enough quality and validity to incorporate these technologies on a high level. As precision agriculture introduces new complexities, it also opens up new future directions in security and other areas.

In this study, we suggested a security framework for the agricultural internet of things that integrates blockchain technology, fog computing, and software-defined networking. The suggested architecture avoids the requirement for simulation-based work with both private and public blockchain for communication between SDN controllers and IoT devices. Here, for IoT connectivity, each SDN controller is linked to a blockchain. SDN controllers are interconnected by a network's topology. One of the key objectives of the proposed architecture is to enhance communication security for IoT devices. For IoT devices with limited resources, distributed trust-based verification makes blockchain even more useful. With a focus on fixing security and data reliability issues in a distributed controller context. Additionally, we want to increase the accuracy of our system using machine learning methods.

It might be significant to add an intrusion detection system utilizing various deep learning algorithms in future work to prevent the insertion of fake Sensor data in the intelligent agricultural field.

**Author Contributions:** Conceptualization, S.P., S.D. and S.R.; methodology, S.P., M.A. (Majed Alowaidi), S.D. and P.P.M.; software, S.P., S.R., M.A. (Majed Alowaidi), P.P.M. and H.A.; validation, M.A. (Mohamed Alshehri), S.R. and H.A.; formal analysis, S.P., S.D., M.A. (Mohamed Alshehri) and P.P.M.; writing—original draft preparation, S.P., S.D. and P.P.M.; writing—review and editing, M.A. (Mohamed Alshehri), S.R. and H.A. All authors have read and agreed to the published version of the manuscript.

**Funding:** The author would like to thank Deanship of Scientific Research at Majmaah University for supporting this work under Project Number No. R-2023-67.

**Institutional Review Board Statement:** Not applicable.

**Informed Consent Statement:** Not applicable.

**Data Availability Statement:** Research data are not shared.

**Conflicts of Interest:** The authors declare that they have no conflict of interest.

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
