# Peer review of "AgriSecure: A Fog Computing-Based Security Framework for Agriculture 4.0 via Blockchain"

_processes, doi:10.3390/pr11030757_

Round 1

Reviewer 1 Report

 This study focused on layered architectural design, identified security issues, and presented security demands and upcoming prospects. In addition to that, authors propose a security architectural framework for the agriculture 4.0 that combines blockchain technology, fog computing, and software-defined networking. The suggested framework combines Ethereum blockchain and software-defined networking technologies on an open-source IoT platform. It is then tested with three different cases under DDoS attack. The results of the performance 32 analysis shows that overall, the proposed security framework has performed well. 

The work is important and timely, however lacks significant merits.

a) It is important to understand the threat model of the work. Since security is a concern, it is critical to know what is the threat.

b) Further, It is important to highlight why the threats are significant and different from other edge frameworks. 

c) Several of the critical works in security of edge based infrasturctures in CPS is missing in literature review including the i) Authorization framework for secure cloud assisted connected cars and vehicular internet of things; (ii) Secure V2V and V2I Communication in Intelligent Transportation using Cloudlets (iii) Cyber attacks on smart farming infrastructure (iv) Towards Activity-Centric Access Control for Smart Collaborative Ecosystems

d) Finally, the deployement architecture is incomplete and lacks significant details. Results need to be compared with other existing frameworks. 

Author Response

Comments and Suggestions for Authors

This study focused on layered architectural design, identified security issues, and presented security demands and upcoming prospects. In addition to that, authors propose a security architectural framework for the agriculture 4.0 that combines blockchain technology, fog computing, and software-defined networking. The suggested framework combines Ethereum blockchain and software-defined networking technologies on an open-source IoT platform. It is then tested with three different cases under DDoS attack. The results of the performance 32 analysis shows that overall, the proposed security framework has performed well. 

The work is important and timely, however lacks significant merits.

  1. a) It is important to understand the threat model of the work. Since security is a concern, it is critical to know what is the threat.

Ans:- As the study is concerned with security challenges in agriculture 4.0, different security aspects have been studied. Section 3 describes the different types of security threats at multilayered paradigm.

.b) Further, It is important to highlight why the threats are significant and different from other edge frameworks. 

Ans:- The significance of threats is highlighted in the introduction section, as per your suggestion. This study focused on fog computing based IoT framework.

  1. c) Several of the critical works in security of edge based infrasturctures in CPS is missing in literature review including the i) Authorization framework for secure cloud assisted connected cars and vehicular internet of things; (ii) Secure V2V and V2I Communication in Intelligent Transportation using Cloudlets (iii) Cyber attacks on smart farming infrastructure (iv) Towards Activity-Centric Access Control for Smart Collaborative Ecosystems

Ans:- In this article, we focused on fog computing-based IoT framework. As you suggested we will definitely focus on the future work towards our study.

  1. d) Finally, the deployment architecture is incomplete and lacks significant details. Results need to be compared with other existing frameworks. 

Ans:- The data analysis with more fieldwork has been carried out in this study and few simulations has been made. One more simulation added in figure 7 as per the suggestion. The comparison state-of-

Reviewer 2 Report

It began with a distinguished introduction that contrasted the old farming environment with modern farming methods employing advanced technologies such as the Internet of Things and agricultural intelligence, and the most significant potential security issues were highlighted in a clear and engaging manner.

· The model implementation environment is presented in the introduction, and the established testing method is explained simply and smoothly as follows. An open-source Internet of Things platform based on the Ethereum blockchain and software-defined networking. Three Attacks put it to the test. A performance analysis demonstrates that the proposed security system addressed them reasonably.

· The language is straightforward, and the abbreviations for all of the terminology are neatly displayed, and it serves the content and context of the sentences.

· A unique and eye-catching portrayal of the process of evolution of agriculture and its concept from the first to the fourth generation, backed by photos and presented in the form of a clear graph.

· Page 5, from lines 182 to 185, sentences need to end with a full stop.

· Previous research that are comparable to the present study are linked in a table that summarizes the studies in a clear chronological order and provides a concise description of each investigation.

· The information has been reinforced with clear, high-quality visual graphics, and it has been presented in a manner that is proportional to the text. This contributes to the consolidation of the idea.

· In order to prevent the information from becoming unclear, it should have been sectioned off into internal chunks in Table 4, each of which should have been separated by a line.

A magnificent and unique subject matter in terms of the manner in which it is presented, a noteworthy quality in the manner in which the issue is presented, and the provision of an enlightening experience for keeping track of knowledge and articulating it in an understandable manner.

Author Response

Comments and Suggestions for Authors

It began with a distinguished introduction that contrasted the old farming environment with modern farming methods employing advanced technologies such as the Internet of Things and agricultural intelligence, and the most significant potential security issues were highlighted in a clear and engaging manner.

Response:- Thank you.

  • The model implementation environment is presented in the introduction, and the established testing method is explained simply and smoothly as follows. An open-source Internet of Things platform based on the Ethereum blockchain and software-defined networking. Three Attacks put it to the test. A performance analysis demonstrates that the proposed security system addressed them reasonably.

Response:- Thank you.

  • The language is straightforward, and the abbreviations for all of the terminology are neatly displayed, and it serves the content and context of the sentences.

Response:- Thank you.

  • A unique and eye-catching portrayal of the process of evolution of agriculture and its concept from the first to the fourth generation, backed by photos and presented in the form of a clear graph.

Response:- Thank you.

  • Page 5, from lines 182 to 185, sentences need to end with a full stop.

Response: - Checked and Corrected.

  • Previous research that are comparable to the present study are linked in a table that summarizes the studies in a clear chronological order and provides a concise description of each investigation.

Response: - Thank you.

  • The information has been reinforced with clear, high-quality visual graphics, and it has been presented in a manner that is proportional to the text. This contributes to the consolidation of the idea.

Response:- Thank you.

  • In order to prevent the information from becoming unclear, it should have been sectioned off into internal chunks in Table 4, each of which should have been separated by a line.

Response:- Checked and Updated.

A magnificent and unique subject matter in terms of the manner in which it is presented, a noteworthy quality in the manner in which the issue is presented, and the provision of an enlightening experience for keeping track of knowledge and articulating it in an understandable manner.

Response:- Thank you.

Reviewer 3 Report

The manuscript addressed Agriculture 4.0 and potential security threats in multi-layered paradigm. It is well-written manuscript involving multiple references, and it can be the significant article in the field of Agriculture and Smart Farming. Thus, this reviewer would like to recommend its publication in Processes as it is. 

Author Response

Comments and Suggestions for Authors

The manuscript addressed Agriculture 4.0 and potential security threats in multi-layered paradigm. It is well-written manuscript involving multiple references, and it can be the significant article in the field of Agriculture and Smart Farming. Thus, this reviewer would like to recommend its publication in Processes as it is. 

Response:- Thank you.

Reviewer 4 Report

The authors of the paper describe their proposed approach for A fog computing-based security framework for agriculture 4.0 via blockchain. The topic is interesting and with possible applicability. However, the paper needs several improvements:

1) the main contribution and originality should be explained in more detail, which part of the proposal is new?

2) the motivation of the approach with needs further clarification, why this work was undertaken?

3) discussion of related work should be expanded with more recent work

4) Minor grammar and syntax issues need correction to enhance readability

5) more simulation results and formal comparison of results are needed to really validate the proposed ideas

6) the conclusions should be extended with more discussion of future works

7) More references to recent related papers could be included

8) There are some typos, like there are two references with a number 1 in the list of references

9) Section 1 is too long, in my opinion the theory should be moved to another section

10) There are no equations in the paper, which can be viewed as possible lack of formality

Author Response

Comments and Suggestions for Authors

The authors of the paper describe their proposed approach for “A fog computing-based security framework for agriculture 4.0 via blockchain”. The topic is interesting and with possible applicability. However, the paper needs several improvements:

1) The main contribution and originality should be explained in more detail, which part of the proposal is new?

Ans: Fog computing via Blockchain technology in the field of Agricultural security is the originality of the study. Following are few contributions:

  • A blockchain-based authenticity monitoring technique to prevent erroneous control and information delivery
  • We presented an SDN-compatible simulated switch to enhance network management.
  • We demonstrate the effectiveness of the suggested security architecture using experimental data from various case studies from an open-source IoT platform incorporating Ethereum blockchain and SDN technologies.

2) The motivation of the approach with needs further clarification, why this work was undertaken?

  • To fulfill the demand on the increasing population and scarcity of agricultural land, new and advanced technology and resources were adopted for increase the productivity in agricultural sectors. Hence the security of the technology is a biggest challenge in adaptation of smart agriculture. This motivated us to undertake a work towards security challenges in this field.

3) Discussion of related work should be expanded with more recent work

  • We have referred to many recent articles in the year 2022 and 2023 related to our study in each section of the article.

4) Minor grammar and syntax issues need correction to enhance readability

Ans. Corrected as per the suggestion

5) More simulation results and formal comparison of results are needed to really validate the proposed ideas

Ans: The data analysis with more fieldwork has been carried out in this study and few simulation has been made. One more simulation added in figure 7.

6) The conclusions should be extended with more discussion of future works

The suggested architecture avoids the requirement for simulation-based work with both private and public blockchain for communication between SDN controllers and IoT devices. Here, for IoT connectivity, each SDN controller is linked to a blockchain. SDN controllers are interconnected by a network's topology. One of the key objectives of the proposed architecture is to enhance communication security for IoT devices. For IoT devices with limited resources, distributed trust-based verification makes blockchain even more useful. With a focus on fixing security and data reliability issues in a distributed controller context. Additionally, we want to increase the accuracy of our system using machine learning methods.

7) More references to recent related papers could be included

Ans: 2 more reference have been added as per the suggestion.

  1. Okolie, C. C., Danso-Abbeam, G., Groupson-Paul, O., & Ogundeji, A. A. (2023). Climate-Smart Agriculture Amidst Climate Change to Enhance Agricultural Production: A Bibliometric Analysis. Land12(1), 50.
  2. Selbonne, S., Guindé, L., Causeret, F., Bajazet, T., Desfontaines, L., Duval, M., ... & Blazy, J. M. (2023). Co-Design and Experimentation of a Prototype of Agroecological Micro-Farm Meeting the Objectives Set by Climate-Smart Agriculture. Agriculture13(1), 159.

8) There are some typos, like there are two references with a number 1 in the list of references.

Ans: Thoroughly checked and corrected.

9) Section 1 is too long, in my opinion the theory should be moved to another section

Ans: As section 1 focuses on the representation of every aspect of smart agriculture along with a brief study, hence it looks lengthy. These aspects were elaborated in different sub-sections.

10) There are no equations in the paper, which can be viewed as a possible lack of formality

Ans: We have only focused on the Simulation and theoretical aspects of the study.

Round 2

Reviewer 1 Report

My initial comments are still unanswered. I suggest authors to review the comments carefully to address them. 

Author Response

Comments and Suggestions for Authors

My initial comments are still unanswered. I suggest authors to review the comments carefully to address them.

Response: Authors are thankful to the reviewer for this very useful observation. As per the suggestions of the reviewer, we now review the comments in the 1st round of revision and revised the paper as per the comments. The point-to-point response to the comments of the 1st round of revision is as follows.

Comments during 1st round of revision

Comments and Suggestions for Authors

This study focused on layered architectural design, identified security issues, and presented security demands and upcoming prospects. In addition to that, authors propose a security architectural framework for the agriculture 4.0 that combines blockchain technology, fog computing, and software-defined networking. The suggested framework combines Ethereum blockchain and software-defined networking technologies on an open-source IoT platform. It is then tested with three different cases under DDoS attack. The results of the performance 32 analysis shows that overall, the proposed security framework has performed well. 

The work is important and timely, however lacks significant merits.

Reviewer#1, Concern # 1: It is important to understand the threat model of the work. Since security is a concern, it is critical to know what the threat is.

Author response: Authors are thankful to the reviewer for this very useful observation. Authors would like to convey that in present work, as the study is concerned with security challenges in agriculture 4.0, different security aspects have been studied. Section 3 describes the different types of security threats at multilayered paradigm.

We have done the simulation study by applying different types of possible threats in different layers of IoT for Agriculture 4.0. Following are the expected threats and its consequences.

We have done the simulation study by applying different types of possible threats in different layers of IoT for Agriculture 4.0. Following are the expected threats and their consequences.

The Physical layer, as is primarily concerned with physical equipment such as sensors and actuators. Physical devices can fail due to unintentional or intentional human behavior, viruses, spyware, or cyber-attacks. This includes different expected threats such as Random device incidents, Mobile Hijacking, Fake nodes, Abnormal Measurement, Sleeplessness attacks, etc.

In Network Layer, The objective of this layer is to send the generated data by the sensors from the physical layer to the most trustable computational unit, which is the cloud. The most common attacks, affect the resources used in the network layer. The security risks in network layers are Distributed Denial of Service, Attacks on data transit, Attacks on routers, malware infusion attacks, Botnets, etc.

In the edge layer, the edge contains important aspects that monitor and control modules communicate with all layers and access resources. The following are major edge security issues. It includes Actuator control forge, Booting attack, Signature wrapping, Measure injection Forgery, Unauthorized access, Gateway-cloud fabrication, etc.

The Application layer, this layer provides end-user assistance and data to process and make system decisions. This layer’s security concerns are application-specific, focusing on preventing data theft and ensuring privacy. Some of the attacks that might affect cloud services and applications are listed below in terms of security such as malicious scripts, Phishing, Denial of service, etc.

Author action: Authors have tried to provide their best clarification to the query of honorable reviewer.

Reviewer#1, Concern # 2: Further, It is important to highlight why the threats are significant and different from other edge frameworks. 

Author response: Authors are thankful to the reviewer for this very useful observation. The biggest obstacle to the growth of cloud computing is security worries. Information security continues to have an impact on the market even though it raises the inconveniences related to information security. The threat of security breaches in the cloud infrastructure must be understood by clients. Cloud- computing capacity, net-working, and computing powers are brought to the edge by fog computing. One of the most critical challenges with fog computing platforms is safety and security. Light-weight devices have benefited greatly from edge computing's ability to complete difficult tasks quickly, but the technology's hasty development has led to a general disregard for safety risks in edge computing stages and the applications they allow.

Author action: Authors have tried to provide their best clarification to the query of honorable reviewer. The updates are highlighted in yellow color in Section 5.

Reviewer#1, Concern # 3: Several of the critical works in security of edge based infrastructures in CPS is missing in literature review including the i) Authorization framework for secure cloud assisted connected cars and vehicular internet of things; (ii) Secure V2V and V2I Communication in Intelligent Transportation using Cloudlets (iii) Cyber-attacks on smart farming infrastructure (iv) Towards Activity-Centric Access Control for Smart Collaborative Ecosystems.

Author response: Authors would like to thank the reviewer for his constructive suggestion. As per the suggestion, the followings articles have been reviewed and included in the Existing Research on Security in Agriculture 4.0 (Section 5).

  1. Sontowski, Sina & Gupta, Maanak & Sree, Sai & Abdelsalam, Mahmoud & Mittal, Sudip & Joshi, Anupam & Sandhu, Ravi. (2020). Cyber Attacks on Smart Farming Infrastructure. 10.1109/CIC50333.2020.00025.
  2. Gupta, Maanak & Benson, James & Patwa, Farhan & Sandhu, Ravi. (2020). Secure V2V and V2I Communication in Intelligent Transportation Using Cloudlets. IEEE Transactions on Services Computing. PP. 1939-1374. 10.1109/TSC.2020.3025993.
  3. Gupta, Maanak & Sandhu, Ravi. (2018). Authorization Framework for Secure Cloud Assisted Connected Cars and Vehicular Internet of Things. 10.1145/3205977.3205994.
  4. Gupta, M., & Sandhu, R.S. (2021). Towards Activity-Centric Access Control for Smart Collaborative Ecosystems. Proceedings of the 26th ACM Symposium on Access Control Models and Technologies.

Author action: Authors have added the articles in the Existing Research on Security in Agriculture 4.0 (Section 5) as per the honorable reviewer’s suggestion in the revised version of the manuscript. All the changes are highlighted in yellow color in the revised manuscript.

Reviewer#1, Concern # 4: Finally, the deployment architecture is incomplete and lacks significant details. Results need to be compared with other existing frameworks. 

Author response: Authors are thankful to the reviewer for this very useful observation. Authors are pleased to inform you that necessary rectification have been made in the experimental setup architecture (Figure 6). Author would like to convey to the honorable reviewer that conceptually the proposed approach is very first of its type as we have considered DDoS attack at the application layer of fog-based IoT framework for agriculture 4.0. A state of art with technological comparison is presented in table 5 of section 5.

Author action: Authors have made necessary rectification have been made in the experimental setup architecture (Figure 6).

Reviewer 4 Report

The paper has been improved, but not all of my previous comments were addressed.

Author Response

Comments and Suggestions for Authors

The paper has been improved, but not all of my previous comments were addressed.

Response: Authors are thankful to the reviewer for this very useful observation. As per the suggestions of the reviewer, we now review the comments in the 1st round of revision and revised the paper as per the comments. The point-to-point response to the comments of the 1st round of revision is as follows.

Comments during 1st round of revision

Comments and Suggestions for Authors

The authors of the paper describe their proposed approach for “A fog computing-based security framework for agriculture 4.0 via blockchain”. The topic is interesting and with possible applicability. However, the paper needs several improvements:

Reviewer#2, Concern # 1:  The main contribution and originality should be explained in more detail, which part of the proposal is new?

Author response: Authors would like to thank the reviewer for his constructive suggestion. Fog computing via Blockchain technology in the field of Agricultural security is the originality of the study. Following are few contributions:

  • A blockchain-based authenticity monitoring technique to prevent erroneous control and information delivery.
  • We presented an SDN-compatible simulated switch to enhance network management.
  • By using Hping tool, we launched DDoS attack on Blockchain network.
  • We demonstrate the effectiveness of the suggested security architecture using experimental data from various case studies from an open-source IoT platform incorporating Ethereum blockchain and SDN technologies.

Author action: Authors have tried to provide their best clarification to the query of honorable reviewer.

Reviewer#2, Concern # 2: The motivation of the approach with needs further clarification, why this work was undertaken?

Author response: Author would like to convey to the honorable reviewer with clarification that this work was done to fulfill the demand on the increasing population and scarcity of agricultural land, new and advanced technology and resources were adopted for increase the productivity in agricultural sectors. Hence the security of the technology is a biggest challenge in adaptation of smart agriculture. This motivated us to undertake a work towards security challenges in this field. The following are key factors which inspired the authors.

  • Smart agriculture is a new paradigm that integrates information technology with conventional farming as a result of the low productivity of traditional agriculture and the extensive usage of information technology. It has the potential to become the next big thing in agricultural development. Consequently, it is crucial to out-line the current manufacturing model and particular studies [27].
  • Despite substantial research on smart agriculture, less has been conducted in comparison to industrial security solutions to analyze security problems.
  • It is crucial to examine the features of security concerns in relation to situations involving smart agriculture [28]. This article attempts to give a review of the se-curity challenges raised by smart agriculture in light of the aforementioned varia-bles, which inevitably results in a significant number of open research questions.

Author action: Authors have tried to provide their best clarification to the query of honorable reviewer.

Reviewer#2, Concern # 3: Discussion of related work should be expanded with more recent work

Author response: Authors would like to thank the reviewer for his constructive suggestion. Authors referred to many recent articles in the year 2022 and 2023 related to our study in each section of the article. Some of them have been included in the revised manuscript.

Author action: Authors added necessary articles in reference section and highlighted in yellow color.

Reviewer#2, Concern # 4: Minor grammar and syntax issues need correction to enhance readability

Author response:  Thank you for the valuable suggestion. The entire manuscript has been meticulously revised against all grammatical errors. All the spellings conform to that of the Oxford English dictionary.

Author action: All the changes are highlighted in yellow color in the revised manuscript.

Reviewer#2, Concern # 5: More simulation results and formal comparison of results are needed to really validate the proposed ideas.

Author response: The authors would like to thank the honorable reviewer for his constructive comments and suggestions. The data analysis with more fieldwork has been carried out in this study and few simulation has been made. One more simulation added in figure 7.

Author action: Authors added one more simulation work in figure 7.

Reviewer#2, Concern # 6: The conclusions should be extended with more discussion of future works

Author response: The authors would like to thank the honorable reviewer for his constructive comments and suggestions. The suggested architecture avoids the requirement for simulation-based work with both private and public blockchain for communication between SDN controllers and IoT devices. Here, for IoT connectivity, each SDN controller is linked to a blockchain. SDN controllers are interconnected by a network's topology. One of the key objectives of the proposed architecture is to enhance communication security for IoT devices. For IoT devices with limited resources, distributed trust-based verification makes blockchain even more useful. With a focus on fixing security and data reliability issues in a distributed controller context. Additionally, we want to increase the accuracy of our system using machine learning methods.

Author Action: Authors updated the conclusion section as per the reviewer’s suggestion in the revised version of the manuscript. All the changes are highlighted in yellow colour in the ‘conclusion’ section of the revised manuscript.

Reviewer#2, Concern # 7: More references to recent related papers could be included

Author response: The authors would like to thank the reviewer for this constructive suggestion. As per the suggestion of honorable reviewer, authors have added most recent references. The details are as follows:

  1. Okolie, C. C., Danso-Abbeam, G., Groupson-Paul, O., & Ogundeji, A. A. (2023). Climate-Smart Agriculture Amidst Climate Change to Enhance Agricultural Production: A Bibliometric Analysis. Land12(1), 50.
  2. Selbonne, S., Guindé, L., Causeret, F., Bajazet, T., Desfontaines, L., Duval, M., ... & Blazy, J. M. (2023). Co-Design and Experimentation of a Prototype of Agroecological Micro-Farm Meeting the Objectives Set by Climate-Smart Agriculture. Agriculture13(1), 159.
  3. Sontowski, Sina & Gupta, Maanak & Sree, Sai & Abdelsalam, Mahmoud & Mittal, Sudip & Joshi, Anupam & Sandhu, Ravi. (2020). Cyber Attacks on Smart Farming Infrastructure. 10.1109/CIC50333.2020.00025.
  4. Gupta, Maanak & Benson, James & Patwa, Farhan & Sandhu, Ravi. (2020). Secure V2V and V2I Communication in Intelligent Transportation Using Cloudlets. IEEE Transactions on Services Computing. PP. 1939-1374. 10.1109/TSC.2020.3025993.
  5. Gupta, Maanak & Sandhu, Ravi. (2018). Authorization Framework for Secure Cloud Assisted Connected Cars and Vehicular Internet of Things. 10.1145/3205977.3205994.
  6. Gupta, M., & Sandhu, R.S. (2021). Towards Activity-Centric Access Control for Smart Collaborative Ecosystems. Proceedings of the 26th ACM Symposium on Access Control Models and Technologies.

Author Action: Authors updated the reference section as per the reviewer’s suggestion in the revised version of the manuscript which are highlighted in yellow color.

Reviewer#2, Concern # 8: There are some typos, like there are two references with a number 1 in the list of references.

Author response: Author response:  Thank you for the valuable suggestion. The entire manuscript has been meticulously revised against all grammatical errors. All the spellings conform to that of the Oxford English dictionary.

Author action: All the changes are highlighted in yellow color in the revised manuscript.

Reviewer#2, Concern # 9: Section 1 is too long, in my opinion the theory should be moved to another section

Author response: The authors would like to thank the reviewer for this constructive suggestion. Necessary changes have been made in the revised manuscript.

Author action: As per the suggestion of honorable reviewer, authors have moved the theory into section 2.

Reviewer#2, Concern # 10: There are no equations in the paper, which can be viewed as a possible lack of formality

Author response: Authors are very thankful to the reviewer for raising this interesting query and would like to convey that as per reviewer’s valuable suggestion, authors tried to include a general mathematical equation for DDoS attack.

Author action: As per the suggestion of honorable reviewer, authors have included necessary mathematical equation (1) which is highlighted in section 3.

Round 3

Reviewer 1 Report

This revision addressed all of my concerns raised in the original manuscript. Overall, there are substantial changes to the original version, and author have careful answered the reviews. It is suggested that the camera -ready manuscript must be thoroughly proof-read.

Reviewer 4 Report

The authors have addressed most of my comments and can now suggest acceptance.